# Targeted Delivery of HSP70 to Tumor Cells via Supramolecular Complex Based on HER2-Specific DARPin9_29 and the Barnase:Barstar Pair

**DOI:** 10.3390/cells13040317

**Published:** 2024-02-09

**Authors:** Ludmila G. Alekseeva, Olga V. Ovsyanikova, Alexey A. Schulga, Maria V. Grechikhina, Olga A. Shustova, Elena I. Kovalenko, Elena V. Svirshchevskaya, Sergey M. Deyev, Alexander M. Sapozhnikov

**Affiliations:** 1M.M. Shemyakin and Y.A. Ovchinnikov Institute of Bioorganic Chemistry, Russian Academy of Sciences, 117997 Moscow, Russia; olgaovsyanickova@yandex.ru (O.V.O.); schulga@gmail.com (A.A.S.); marygrec@mail.ru (M.V.G.); olga_shustova@list.ru (O.A.S.); lenkovalen@mail.ru (E.I.K.); esvir@yandex.ru (E.V.S.); deyev@mail.ibch.ru (S.M.D.); amsap@mail.ru (A.M.S.); 2Faculty of Biology, Lomonosov Moscow State University, 119234 Moscow, Russia

**Keywords:** cancer immunotherapy, tumor-targeted delivery, HSP70, DARPin9_29, HER2/neu, barnase:barstar, NK cells

## Abstract

(1) Background: We have previously shown that the use of an artificial supramolecular two-component system based on chimeric recombinant proteins 4D5scFv-barnase and barstar-heat shock protein 70 KDa (HSP70) allows targeted delivery of HSP70 to the surface of tumor cells bearing HER2/neu antigen. In this work, we studied the possibility to using DARPin9_29-barnase as the first targeting module recognizing HER2/neu-antigen in the HSP70 delivery system. (2) Methods: The effect of the developed systems for HSP70 delivery to human carcinomas SK-BR-3 and BT474 cells hyperexpressing HER2/neu on the activation of cytotoxic effectors of the immune cells was studied in vitro. (3) Results: The results obtained by confocal microscopy and cytofluorimetric analysis confirmed the binding of HSP70 or its fragment HSP70-16 on the surface of the treated cells. In response to the delivery of HSP70 to tumor cells, we observed an increase in the cytolytic activity of different cytotoxic effector immune cells from human peripheral blood. (4) Conclusions: Targeted modification of the tumor cell surface with molecular structures recognized by cytotoxic effectors of the immune system is among new promising approaches to antitumor immunotherapy.

## 1. Introduction

A new promising approach to antitumor immunotherapy is the loading onto the tumor cell surface molecular structures recognized by innate immunity cytotoxic effectors such as natural killers (NK). There is ample evidence that 70 kDa heat shock proteins (HSP70) are such “elimination markers” capable of activating NK cells and other cytotoxic lymphocytes of the immune surveillance system [1]. Tumor cells do not expose HSPs on their surface in a steady-state condition, but they express it under stress such as a heating.

Previously, we developed the two-component system to deliver HSP70 to the tumor cell surface consisting of the chimeric recombinant proteins 4D5scFv-dibarnase and barstar-HSP70 or barstar-HSP70_16, where HSP70_16 is a 16 kDa C-terminal fragment of HSP70 protein [2]. The first targeting module was constructed from a 4D5 antibody fragment (scFv), binding the extracellular domain of HER2/neu receptor [3], and fused with two molecules of a bacterial ribonuclease barnase [4]. The second “effector” module contained barstar, a natural inhibitor of barnase. Barstar and barnase form heterodimers with *K*_D_~10^−14^ M [5], exhibiting extraordinary stability in severe conditions. The specific binding of HSP70 to the HER2/neu-expressing tumor cells in vitro had a significant stimulating effect on the cytotoxic activity of NK cells [2]. 

Recently, scaffold proteins such as monobody, anticalines, affibodies, or darpins have been considered worldwide as an alternative to targeting antibodies [6,7]. In particular, the HER2-specific scaffold protein of non-immunoglobulin nature DARPin9_29 (short for designed ankyrin repeat proteins) has been widely used in recent years as an HER2/neu-targeting molecule instead of the 4D5scFv [8]. DARPin9_29 is an 18 kDa protein with a binding constant to HER2 of around 3.8 × 10^9^ [9]. Due to its relatively small molecular mass and, consequently, lower immunogenicity and higher ability to penetrate tissues, darpins are more attractive for therapeutic usage than antibodies. Darpin molecules are highly stable, and the absence of disulfide bonds allows their production in a bacterial expression system in high yield.

In this work, we developed an HSP70 delivery system using recombinant DARPin9_29-barnase protein as the antigen-recognizing targeting module and barstar-HSP70 or barstar-HSP70_16 proteins as the second one. For the comparison, we also produced and studied the hybrid single-module DARPin9_29-HSP70 protein. 

## 2. Materials and Methods

### 2.1. Construction of DARPin9_29-HSP70 Expression Plasmid

To produce DARPin9_29-HSP70 vector, the DARPin9_29 sequence was amplified from pDARPin9_29-mCherry [10] using 5′-tattccatatggacctgggtaagaaactg and 5′-ctgacagaattcggcgccttgcaggatttcagccag primers. Both DARPin9_29 amplicon and plasmid pET22-His6-barstar-HSP70 [2] were digested with *Nde*I and *Spe*I, and the DARPin9_29 gene was cloned into the pET22 vector in the same reading frame with HSP70. The resulting expression cassette (Figure 1a) consisted of an inducible T7 promotor, hexahistidine tag, and DARPin9_29 and HSP70 genes. The DARPin9_29 and HSP70 were connected by a flexible linker from the mouse IgG3 hinge region [3]. After sequence verification, the plasmid pET22_His6-DARPin9_29-HSP70 was used for the protein expression. The recombinant proteins DARPin9_29-barnase, barstar-HSP70, and barstar-HSP70_16 were produced as described in [2,11].

### 2.2. Expression and Purification of DARPin9_29-HSP70

Freshly transformed *E. coli* BL21 (DE3) cells were grown in SOB medium containing 0.1 g/L ampicillin; the *lac*-promoter was induced with 1 mM ITPG at OD_600_ of 0.7. Expression was allowed to continue for 12 h at 37 °C. Cells were harvested by centrifugation at 6000× *g* for 15 min at 4 °C, and the cell pellet was resuspended in lysis buffer (20 mM Tris-base, 20 mM NaCl, 20 mM MgCl_2_, and 1 mg/mL lysozyme). The suspension was incubated for 30 min at room temperature and then sonicated on ice. Cellular debris was removed by centrifugation at 15,000× *g* for 30 min at 4 °C. 

The solution was applied to a Ni^2+^-NTA column (GE Healthcare) equilibrated in 100 mM NaH_2_PO_4_, 10 mM Tris-HCl, 8 M urea, and pH 8.0 buffer (B1). DARPin9_29-HSP70 was eluted with pH 4.5 B1 buffer. The protein yield was 10 mg/L of growth medium.

### 2.3. PAAG Gel-Electrophoresis

Whole lysate of recombinant *E. coli* and purified DARPin9_29-HSP70 (30 µg/line) was denatured at 95 °C in Laemmli sample buffer for 5 min under reducing conditions (2-mercaptoethanol). Proteins were separated by 10% PAAG gels using a Bio-Rad (Singapore) electrophoresis system. Gels were stained with PageBlue™ Protein Staining Solution (Thermo Scientific, Waltham, MA, USA). Molecular weight (MW) standard PageRuler™ unstained Protein Ladder, 20 to 250 kDa (Thermo Scientific, Waltham, MA, USA), was used to verify the MW of the proteins.

### 2.4. Cells

Human HER2-positive adenocarcinoma cells SK-BR-3 and BT-474 were grown in RPMI-1640, supplemented with 10% fetal calf serum (FCS) and pen-strep-glut (all from PanEco, Moscow, Russia) (complete culture medium) in a CO_2_ incubator at 37 °C. Adhesive cells were passaged using Trypsin /EDTA solution (PanEco, Moscow, Russia) twice a week.

### 2.5. Interaction of the Dual-Module Molecular Constructs with HER2-Positive Cells

The SK-BR-3 and BT-474 cells were detached using Versen solution and transferred to a phosphate–salt buffer (PBS) containing 1% BSA and 0.1% NaN3, pH 7.4 (PBA). The 3 × 10^5^ cells were then incubated with 100 μL of DARPin9_29-barnase (20 μg/mL, 40 min), washed twice with PBS and incubated with 100 μL of barstar-HSP70 or barstar-HSP70-16 (50 μg/mL, 40 min at +4 °C) and washed again. The single-module DARPin9_29-HSP70 was used at a concentration of 50 μg/mL. The negative control was treated in the same way but contained no constructs.

### 2.6. Flow Cytometry

Cell lines BT-474 and SK-BR-3 were pretreated with the constructs DARPin9_29-barnase/barstar-HSP70 or DARPin9_29-barnase/barstar-HSP70-16, washed and labeled with the antibody to HSP70, BRM-22 at 10 μg/mL, followed by the second antibody to murine IgG-FITC (both from Sigma, Merck KGaA, Darmstadt, Germany) at a dilution of 1:1000 in PBA for 40 min at +4 °C. The cells were analyzed by a FACSCalibur cytometer (BD, Franklin Lakes, NJ, USA) using propidium iodide to exclude dead cells. Data were analyzed by the FlowJo program, version 10 (BD, Franklin Lakes, NJ, USA).

### 2.7. Confocal Microscopy

SK-BR-3 and BT-474 cells were seeded on sterile cover glasses (15 × 10^4^) placed into 6-well plates (Costar, Washington, DC, USA) and incubated overnight. Afterward, the cells were washed in PBA and incubated with DARPin9_29-barnase (20 μg/mL 40 min) and barstar-HSP70 or barstar-HSP70-16 (50 μg/mL, 40 min) and DARPin9_29-HSP70 (50 μg/mL, 40 min) followed by BRM-22 and anti-mouse IgG-Alexa Fluor 488 (Molecular Probes, Eugene, OR, USA) for 40 min at +4 °C. All the components were introduced sequentially with the wash in-between, stained with nuclear dye Hoechst 33,342 (Sigma, Merck KGaA, Darmstadt, Germany) and, finally, fixed with 2% paraformaldehyde and polymerized with Mowiol 4.88 medium (Calbiochem, Darmstadt, Germany). Slides were analyzed using an Eclipse TE2000 confocal microscope (Nikon, Tokyo, Japan). EZ-C1, version 3.90 FreeViewer software (Nikon Corporation, Tokyo, Japan) was used to process the results.

### 2.8. Isolation of Human Blood Mononuclear Cells

Peripheral blood mononuclear cells (PBMC) were obtained from the blood of healthy anonymous volunteers who gave their informed consent for participation in the study. All studies involving human cells were conducted in accordance with the guidelines of the World Medical Association’s Declaration of Helsinki. According to the national regulations, no additional approval by the local ethics committee was required in the case of anonymous blood cells being discarded after the experiment. PBMCs were isolated by density gradient centrifugation. The cells were then washed twice with Dulbecco PBS (DPBS) for 15 min at 300 g. The isolated PBMCs were then resuspended in magnetic separation buffer (DPBS, 2 mM EDTA, and 0.5% BSA).

### 2.9. NK Cell Isolation

NK cells were isolated by negative magnetic separation from the PBMCs using a MACS NK Cell Isolation Kit (Miltenyi Biotec, Bergisch Gladbach, Germany). After isolation, NK cells were transferred into complete culture medium and incubated overnight in a CO_2_ incubator at 37 °C. Human recombinant interleukin 2 (IL-2) (NPK BioTech, Sankt-Peterburg, Russia) was added to NK cells (500 U/mL) for 12 h before the experiments. A FACSVantage DiVa cell sorter (BD, Franklin Lakes, NJ, USA) was used to sort CD57^+^ and CD57^−^ NK cell subsets. Sorted cells were transferred into culture medium and used in the experiments.

### 2.10. γδT Lymphocyte Isolation

Human γδT lymphocytes were isolated from PBMCs by positive magnetic separation using a gamma/delta T cell kit (MACS Mitenyi, Waltham, MA, USA). After separation, the cells were cultured for 6 to 7 days in complete culture medium supplemented with 10 ng/mL IL-2 and IL-4.

### 2.11. Lactate Dehydrogenase Cytotoxic Test

The CytoTox96 non-radioactive cytotoxicity assay (Promega, Fitchburg, MA, USA) was used to analyze the cytotoxic activity of NK cells. For this assay, SK-BR-3 cells were grown in a 96-well plate overnight (10^4^/well). The cells were treated in PBS with the different modules as described above and finally transferred to complete culture medium. PBMCs or γδT cells were added to the treated cells at 10:1, 20:1 (PBMCs), and 5:1 (γδT cells) ratios. The plates were centrifugated to assist cell contacts and incubated in the CO_2_ incubator for 3 h. During the incubation, a lysing buffer was added to some wells 45 min before the end of the process as a control. At the end of the incubation, the plates were centrifuged, and the supernatant was transferred to another plate with the substrate mixture containing lactate, diaphorase, tetrazolium salts, and NAD+. The plates were incubated for 30 min. The concentration of lactate dehydrogenase was determined by converting tetrazolium salts into formazan. Cytotoxicity was analyzed as the ratio of the optical densities of the samples with a mixture of SK-BR-3 and NK cells to the sample with fully lysed targets, taking into account corrections for the presence of lactate dehydrogenase in the medium and the spontaneous release of lactate dehydrogenase from NK cells and SK-BR-3 cells. Optical density was measured using a Multiscan FC (Thermo Scientific) at 490 nm.

### 2.12. NK Cell Degranulation Cytotoxicity Test

Cytotoxicity was analyzed by assessing the expression level of past degranulation marker LAMP-1 (CD107a) on the surface of NK cells. NK cells were added at a 10:1 ratio to the SK-BR-3 cells treated before with the recombinant proteins as described above. Mouse anti-human monoclonal CD107a-APC (clone REA792, Miltenyi Biotec) antibody diluted in PBA (10 μg/mL) and brefeldin A (10 μg/mL) was added to stain NK cells and prevent LAMP-1 reuptake accordingly. The plate was centrifuged at 450× *g* for 2 min to assist cell contacts and incubated in 5% CO_2_ at 37 °C for 2.5 h. The plate was then centrifuged and washed twice with PBS. Analysis was performed using a FACSCalibur flow cytometer (Becton Dickinson, San Jose, CA, USA). Data processing and statistical analysis were performed in the FlowJo program, version 10 (BD, Franklin Lakes, NJ, USA).

### 2.13. Statistical Processing of the Obtained Data

In this study, the paired Student’s *t*-test was used for statistical processing of the results. The results of five independent experiments were evaluated with a statistical significance level of *p* < 0.05.

## 3. Results

### 3.1. Recombinant Protein DARPin9_29-HSP70

The structure of DARPin9_29-HSP70 hybrid protein is shown in Figure 1. It contains a guide functional module HER2-specific DARPin9_29 [9] and the hexahistidine tail at its N-terminus. HSP70 was attached to the C-terminus of DARPin9_29 via the flexible peptide linker TPLGDTTHTSG, derived from the hinge site of mouse IgG3. The target protein gene was expressed in the *E. coli* BL21(DE3) strain. DARPin9_29-HSP70 fusion protein was purified by metal-chelate affinity chromatography.

### 3.2. Visualization of HSP70 on Tumor Cell Membranes

Both SK-BR-3 and BT474 cell lines highly express the HER2/neu oncomarker (Appendix A and Appendix A). At the same time, they do not expose HSP70 on their surface (Appendix A). To deliver HSP70 to the cell surface, a single-module DARPin9_29-HSP70 or two-component systems capable of forming supramolecular complexes DARPin9_29-barnase:barstar-HSP70 and DARPin9_29-barnase:barstar-HSP70_16 were used to treat the cells. For this, SK-BR-3 and BT474 cells were sequentially incubated with the first module of DARPin9_29-barnase followed by the second module of barstar-HSP70 or barstar-HSP70-16, as well as with a single-module construct of DARPin9_29-HSP70, washed and then stained with BRM-22 antibodies specific to the C-terminus of HSP70 to detect surface-exposed HSP70. 

The confocal microscopy results showed that the DARPin9_29-HSP70 bound BT474 cells more intensely than SR-BR-3 cells (Figure 2a,d). Both two-component constructs effectively delivered HSP70 to the cell surface of both cell lines (Figure 2b,c,e,f). This effect was specific, as the first or the second modules alone or the anti-HSP70 antibody alone have not stained the cell surface (Appendix A). The lower binding efficacy of DARPin9_29-HSP70 hypothetically can be explained by a lower affinity for the HER2/neu receptor caused by a steric hindrance posed by the massive HSP70 molecule, resulting in a reduced number of HSP70 molecules on the surface of the target cells.

Quantitative analysis of HP70 or HSP70_16 expression on the cells was performed by flow cytometry. The results correlated well with the confocal microscopy data; however, the DARPin9_29-barnase:barstar-HSP70_16 effect was better than that of DARPin9_29-barnase:barstar-HSP70, the longest construct (Figure 3). These data support a negative role for the massive HSP70 molecule decreasing the retention of the complex on the cell surface. Consequently, a shorter DARPin9_29-barnase:barstar-HSP70_16 construct could be more efficient for the in vivo experiments. 

### 3.3. Recognition of HSP70 on the Surface of Tumor Cells by PBMC

Recognition of planted HSP70 by PBMC isolated from the blood of healthy donors (Appendix A) was studied in SK-BR-3 cells incubated with or without IL-2 using a lactate dehydrogenase assay. The cells were pretreated with the constructs and co-incubated with PBMC at 10 to 20 PBMC to SK-BR-3 cell ratios. The major effects were found again for the dual barstar:barnase HSP70 and HSP70_16 complexes without a statistical difference between them (Figure 4). A single DARPin9_29-HSP70 molecule was less effective. Activation of PBMC with IL-2 slightly increased the cytotoxic effect. 

### 3.4. Recognition of HSP70 on the Surface of Tumor Cells by NK Cell Subsets and γδT Cells

It is known that HSP70 on the surface of tumor cells is a trigger factor for CD56^bright^/CD94^+^ natural killer cells [12]. We assume that, in our case, it is the NK cells that exert the main cytotoxic effect against the tumor target cells. CD56^+^ NK cell subsets with high and low CD57 expression were isolated from peripheral blood of healthy donors, activated with IL-2 overnight, and co-cultivated with the target cells treated with the HSP70 constructs. The HPS70 loading only slightly increased the cytotoxic activity of the CD57^−^ subset (Figure 5a), which may be explained by the HSP70-independent response of NK cells in this model.

Since the PBMC fraction is very heterogeneous and contains different types of lymphocytes, we also evaluated the cytotoxic activity of γδT lymphocytes isolated by positive magnetic separation from the PBMC of the peripheral blood of healthy donors. SK-BR-3 cells served as targets in this experiment; the ratio of effectors to target numbers was 1 to 5; cytotoxicity was determined by the lactate dehydrogenase assay.

The effect of the γδT lymphocytes slightly differ from the PBMC ones; namely, the major effect was found in the SK-BR-cells pretreated with the longest construct, DARPin9_29-barnase:barstar-HSP70; less efficient was DARPin9_29-HSP70 (Figure 5). A more precise characterization of the immune response induced by HSP70 on the surface of tumor cells requires additional studies.

## 4. Discussion

HSP70 has long attracted the attention of researchers for use in antitumor therapy, but in the vast majority of cases, they are referring to complexes of HSP70 with tumor-specific peptides, which can be used to form tumor-specific immunity [13,14,15]. We propose a fundamentally different approach that is aimed at the stimulation of the immune surveillance system, primarily at the development of a universal antitumor response of cytotoxic effector cells of innate immunity.

The antitumor response of cytotoxic effector cells of innate immunity (NK cells) associated with the localization of HSP70 on the surface of cancer cells was studied and described in detail by G. Multhoff [16]. This team continues to develop this area of research at present, but in the applied aspect of antitumor immunotherapy, they proposed not the delivery of exogenous HSP70 to the surface of target cells but the use of monoclonal antibodies recognizing membrane-associated HSP70 [17]. Thus, we believe that HSP70 is a promising target for the creation of molecular constructs, designed for modification of the surface of tumor cells to enhance antitumor cytotoxic immune response.

Targeted delivery of “cytolytic markers” can be accomplished by including in the recombinant constructs a guide module to other cancer markers, different from HSP, expressed on the surface of a wide range of malignancies. Experiments with targeted delivery of exogenous HSP70 to tumor-associated antigens are described by Poznanski’s group [18]. In that study, a recombinant protein containing a mini-antibody to mesothelin, which is highly expressed in a mouse model of ovarian tumor, and mycobacterial HSP70 was produced and tested.

In this study, we use an original method of targeted delivery of HSP70, directed to the tumor antigen HER-2/neu, to the surface of tumor cells using a two-component construct capable of forming a supramolecular complex through HER2-DaRPin9_29 and barnase:barstar pair interactions. In this system, the function of the first module is to target binding to the surface of cancer cells. In our previous work [2], we used as a recognizing HER2-receptor module the scFv 4D5-dibarnase, which consists of two barnase molecules that are fused serially to the single-chain variable fragment (scFv) of humanized 4D5 antibody (30 kDa). The present work focuses on DARpin9_29, which is an artificial scaffold protein designed as an alternative to antibodies. The advantages of this so-called alternative binding protein include its small size (18 kDa), which facilitates tumor penetration, the absence of Fc antibody- and complement-mediated cytotoxicity, and the high thermostability that enables its long-term storage at room temperature without loss of activity. The simplicity of production of DARpin9_29 fusion proteins makes them promising for creating bispecific and multivalent constructs. Our confocal microscopy (Figure 2) and flow cytometry data (Figure 3) showed that DARPin9_29 in the dual-module system is as effective as the antibodies. The delivery of both the HSP70 and HSP70_16 molecules with DARPin9_29 also significantly enhanced the antitumor cytolytic effect of PBMC (Figure 4).

The barnase exposed on tumor cells serves as a site of selective binding of the second module consisting of barstar and HSP70 (or a fragment thereof). The selective interaction between the first and second modules is facilitated by the uniquely high binding constant of barstar to barnase; this protein heterodimer forms a complex with a K_D_~10^−14^ M, comparable only to the streptavidin-biotin system (K_D_~10^−14^ M). Our studies demonstrated the effectiveness of the barnase:barstar complex for the delivery of various drugs to cancer cells [2,3,19]. 

The originality of the proposed immunotherapy method is also related to the application of a two-component molecular construct. Two-stage sequential treatment of tumor cells has an undoubted advantage in the delivery of HSP70 to tumor tissues in comparison with a single-module DARPin9_29-HSP70 (Figure 2 and Figure 3). It is known that HSP70 is able to form, using its substrate-binding site, strong complexes with the denatured proteins present, in particular, on the surface of stressed, damaged, dead, and transformed cells [13]. Obviously, there is competition between the guiding part of the construct (mini-antibody, darpin, etc.) and the substrate-binding site in the HSP70 molecule for interaction with different target cells, and this will undoubtedly lead to the decreased efficiency of the targeted delivery of HSP70 to the cancer cell surfaces. The interaction of barstar with barnase is characterized by an association constant that is much higher than the constant for the interaction of HSP70 with the substrate molecules that may be present on the surface of other cells that are not targets of the current therapy. When using a two-component system, it is also possible to regulate the place, time, and number of administered drugs to maximize the therapeutic effect and reduce the number of adverse reactions.

In addition, according to our data, treatment of cells with the single-module DARPin9_29-HSP70 yields a much lower concentration of HSP70 on cell membranes compared to the dual-module system, which is likely a consequence of the lower affinity to the HER2/neu receptor of this chimeric protein due to steric hindrance created by the massive HSP70 molecule (Figure 2 and Figure 3). The same is true when comparing in our delivery system the second modules containing either the full-length HSP70 protein or its C-terminal fragment with a molecular mass of 16 kDa. The maximum fluorescence intensity determined by flow cytometry (Figure 3) and proportional to the number of bound molecules was observed for the 16 kDa fragment of HSP70. However, the full-length protein HSP70-labeled cells are more efficiently lysed by PBMC and isolated γδT lymphocytes (Figure 4 and Figure 5).

We have demonstrated in our models the cytolytic response of cytotoxic immune effectors to membrane-associated HSP70 (Figure 4 and Figure 5). The results of the experiments using PBMC as effector cells demonstrated a reliable anti-cancer effect of the treatment of tumor cells with the two-module construct. This small effect was recorded in experiments with cytotoxic γδT lymphocytes, and a similar trend was observed in experiments with NK cells isolated from peripheral blood. The latter can be explained by the high level of HSP70-independent cytotoxic activity of NK cells against the target tumor cells. As for PBMC, we hypothesize that T lymphocytes may have a cytotoxic effect on tumor cells with membrane-associated HSP70. At the same time, we do not exclude the involvement of antigen-presenting cells contained in PBMCs in the development of the lytic reaction of T lymphocytes. Additionally, antigen-presenting cells can also enhance the cytotoxic response of NK cells by produced cytokines or/and by the interaction of classic and non-classic HLA class I molecules with the NK cell-activating receptors. The exact mechanisms of the cytotoxic action of NK cells and other cytotoxic lymphocytes against target cells carrying such constructs require further studies.

It is undoubtedly important to search for the region of the HSP70 molecule that induces the maximum level of cytolytic response of NK cells and other cytotoxic effectors of the immune system, since the response of cytotoxic cells used in this work differed in its specificity. It follows from published experimental data [20] that the C-terminal domain of HSP70 is responsible for the stimulation of the cytotoxic and proliferative activities of NK cells. However, the results of our studies using a panel of monoclonal antibodies, which contains antibodies selectively interacting with the C- or N-terminal fragment of HSP70 [21], indicate that membrane-associated HSP70 is predominantly recognized on many types of tumor cells by antibodies interacting not with the C- but with the N-terminal part of these proteins. This indicates a C-terminal interaction of the HSP70 with the cell surface, leading to an orientation to the outer space of the N-terminal portion of this molecule. Significantly, such target cells were efficiently recognized and subjected to cytolytic attack by cytotoxic effectors of the immune system [22]. The influence of the spatial orientation of HSP70, as well as the search for regions of the molecule that induce cytotoxicity in different populations of cytotoxic lymphocytes, requires further investigation.

## 5. Conclusions

In the present study, we developed the two-component system for the delivery of HSP70 to the surface of HER2-positive tumor cells. This system is based on chimeric recombinant proteins, where the first antigen-recognizing module DARPin9_29-barnase contained a scaffold protein of DARPin9_29 as a guiding molecule to the HER2/neu receptor and a ribonuclease barnase. The second module contained a specific inhibitor of barnase barstar and HSP70 (barstar-HSP70) or its C-terminal 16 kDa region (barstar-HSP70_16) responsible for stimulation cytotoxicity of innate immunity cells. The specific binding of HSP70 with SK-BR-3 and BT474 tumor cells in vitro increased the cytolytic activity of PBMCs and γδT cells. For the comparison, we produced a single-module chimeric protein, DARPin9_29-HSP70, but it was much less efficient than the dual complex. 

Thus, we assume that the targeted modification of the tumor cell surface with molecular structures that are recognized as “elimination markers” by cytotoxic effectors of the innate immune system is among the new promising approaches to antitumor immunotherapy. The exact mechanisms of cytolytic action of NK cells and other cytotoxic lymphocytes, as well as the search for regions of the HSP70 molecule responsible for this effect, require further studies.

## Figures and Tables

**Figure 1 cells-13-00317-f001:**
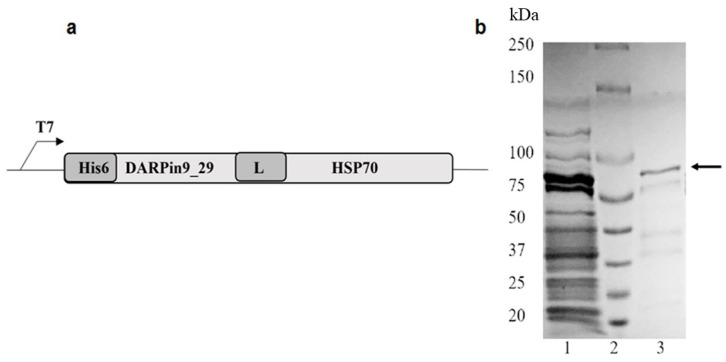
Production of DARPin9_29-HSP70. (**a**) Gene construct for the expression of the DARPin9_29-HSP70: His6, N-terminal 6-His-tag; DARPin9_29, coding region of DARPin9_29; L, hinge-like linker derived from murine IgG3 hinge region; HSP70, coding region of HSP70; (**b**) Coomassie blue-stained SDS-PAGE: whole lysate of *E. coli* strain BL21 after induction with IPTG (Lane 1); molecular weight markers (Lane 2); purified DARPin9_29-HSP70 (Lane 3, arrow).

**Figure 2 cells-13-00317-f002:**
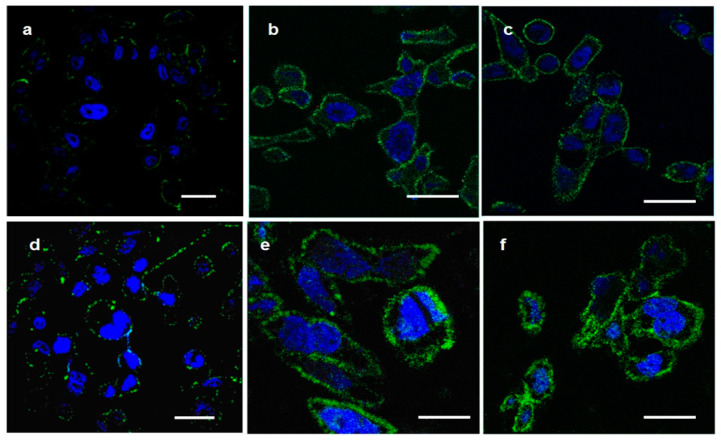
Delivery of HSP70 to the surface of SK-BR-3 (**a**–**c**) or BT474 (**d**–**f**). Cells were treated with unlabeled DARPin9_29-HSP70 (**a**,**d**), DARPin9_29-barnase:barstar-HSP70 (**b**,**e**), or DARPin9_29-barnase:barstar-HSP70_16 (**c**,**f**), washed and stained with anti-HSP70 (green); cell nuclei were stained with Hoechst 33,342 (blue). Scale bars 10 μm.

**Figure 3 cells-13-00317-f003:**
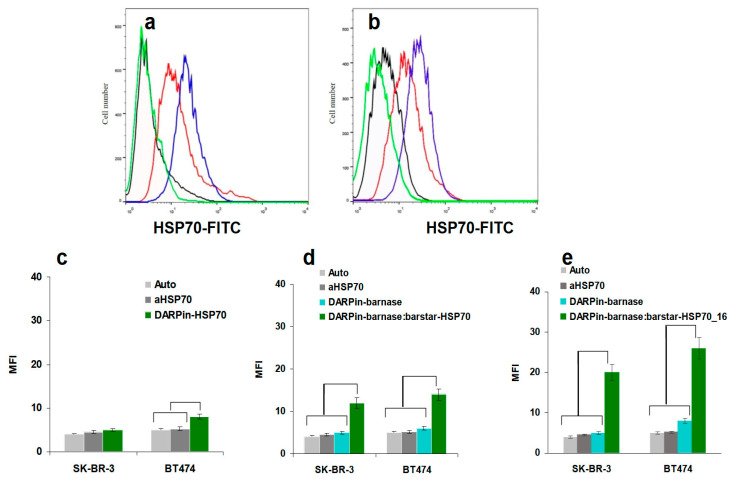
Quantitative analysis of HSP70 delivery to SK-BR-3 or BT474 cell surface. (**a**,**b**): Expression of HSP70 by SK-BR-3 (**a**) or BT474 (**b**) control cells (green lines) or cells treated with a single-module DARPin9_29-HSP70 (black lines), DARPin9_29-barnase:barstar-HSP70 (red lines), and DARPin9_29-barnase:barstar-HSP70_16 (blue lines). The cells were incubated as described in the Section 2. (**c**–**e**) The geometric mean fluorescence intensity of SK-BR-3 and BT474 cells treated with DARPin9_29-HSP70 (**c**), DARPin9_29-barnase:barstar-HSP70 (**d**), and DARPin9_29-barnase:barstar-HSP70_16 (**e**), where auto—autofluorescence and aHSP70—anti-HSP70 BRM22. The data are presented as mean values with standard deviations. The statistical significance (*p* < 0.05 by Students’ *t*-test) is shown with the brackets.

**Figure 4 cells-13-00317-f004:**
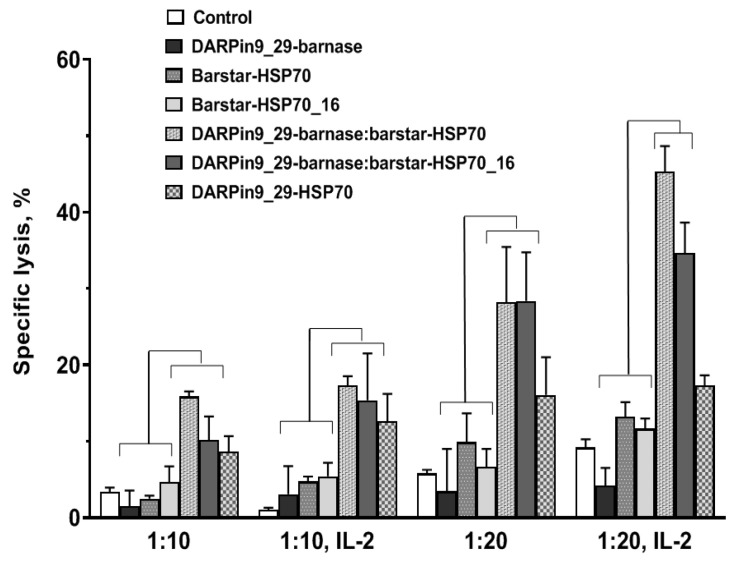
Cytotoxic activity of non- or interleukin-2 activated PBMCs against SK-BR-3 tumor cells planted with HPS70 using the developed constructs. Treated SK-BR-3 cells were incubated with PBMC cells, and cytotoxicity was assessed by the lactate dehydrogenase assay; 1:10; 1:20—ratios of target cells to effectors. The data are presented as mean values with standard deviations. The statistical significance (*p* < 0.05 by Students’ *t*-test) is shown with the brackets.

**Figure 5 cells-13-00317-f005:**
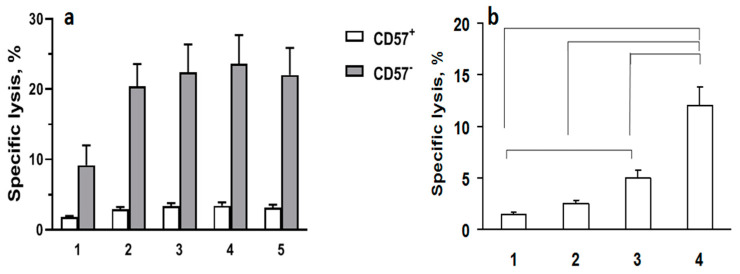
Cytotoxic activity of CD57^+^ and CD57^−^ subsets of NK cells and γδT cells against the treated SK-BR-3: (**a**) cytofluorimetric analysis of degranulation of CD57^+^ and CD57^−^ subsets of NK cells. Treated SK-BR-3 cells were incubated with NK cells, and cytotoxicity was assessed by the expression level of NK cell degranulation marker LAMP-1 (CD107a): 1—spontaneous degranulation; 2—control SK-BR-3 cells without treatment; 3—DARPin9_29-barnase:barstar-HSP70; 4—DARPin9_29-barnase:barstar-HSP70_16; 5—DARPin9_29-HSP70. (**b**) Cytotoxic activity of γδT cells. The treated SK-BR-3 cells were incubated with γδT cells, and cytotoxicity was assessed by the lactate dehydrogenase assay. Ratio of target cells to effectors is 1:5. 1—control SK-BR-3 cells without treatment; 2—DARPin9_29-barnase; 3—DARPin9_29-barnase:barstar-HSP70; 4—DARPin9_29-barnase:barstar-HSP70_16. The data are presented as mean values with standard deviations. The statistical significance (*p* < 0.05 by Students’ *t*-test) is shown with the brackets.

## Data Availability

The data of the study will be available upon request.

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
