# Peer review of "Targeted Delivery of HSP70 to Tumor Cells via Supramolecular Complex Based on HER2-Specific DARPin9_29 and the Barnase:Barstar Pair"

_cells, 2024, doi:10.3390/cells13040317_

Round 1

Reviewer 1 Report

Comments and Suggestions for Authors

The paper “Targeted delivery of HSP70 to tumor cells via supramolecular complex based on Her2-specific DARPin9_29 and the barnase:barstar pair” in cells is a article that  aims to demonstrate the efficiency in delivery HSP70 to the cell surface of the tumor cells (in this case HER2 expressing cells) using the technology Her2-specific DARPin9_29 and the barnase:barstar pair. This would allow the recognition through HSP70 and elimination of the tumor cells by the immune system. The article subject is of interest for the community, however I have several major concerns that lead me to refuse the

the paper for publication.  

One of my major concerns include the description of the results and its discussion that I consider not been clearly described. For instance, it lacks the number of replicates of each experiments,  and importantly the characterization of the PBMCs before and after incubation with the target cells was not done. Thus it is not clear which cells were responsible to the lysis of the target cells and the importance of IL2 it is not clearly mentioned (Figure 4). Furthermore, other major comments can be read below:

Line 23 and below: As already mentioned, please clarify it, as PBMCS also include B , DC cells which do not have a cytotoxic activity...

In material and methods:

-       Line 100: Please indicate the amount of cells?

-       2.5 Paragraph: Please indicate the concentrations used?

-       Please indicate what means PBA?

Results:

The discussion of the results needs to be improved (or in this chapter or in the discussion chapter). The authors do not properly discuss for instance the differences observed between DARPin9_29-HSP70/ DARPin9_29-HSP70:barnase-HSP-70/16.

Figure 1: Can you please explain the appearance of the other below bands, does it means that it is not pure or degraded ? How many milligrams could you obtained.

Figure 3: It would be easier for the reader to have the legend at the side of the histograms/ graphs. Considering that there is space to include the legends, please correct it.

Same comment for the below figures.

Please clarify what is fluorescence intensity (it is a FITC) in a and b.

In c , by fluorescence intensity, you mean geometric mean ? Please clarify it. Same comment for the below graph/histograms.

Which statistic test was used, please indicate it here.

What means autofluorescence, it is from no stained cells?

Paragraph 3.4: To better understand the effect of the combination in the induction of the cytotoxicity of the target cells, I strongly suggest that the cells population will be characterized (type of cells ) for instance using specific antibody staining (e.g. CD20, CD3, CD14, CD16, CD45 etc)of the cells and analysed by flow cytometry.

I do not agreed in using this term as PBMCs as it includes several types if cells including non cytotoxic cells. Moreover the authors should characterized the PBMCs upon stimulation with IL2, meaning type of cells that were expanded.

Line 334: please indicate for how long they were incubated with IL2 before used as effector cells, and thus characterize the type of cells obtained after stimulation.

Figure 4: Consider the below comments. Please indicated/explained why those two ratios were done and better explain the difference observed between the different groups.

Line 352- it is not observed the same effect for the HSP70 alone?

Paragraph 3.5/3.6: I could not fully understand why the authors decided to test the effect of these combinations using NK cells or γδT cells and nor for instance CD4, CD8 T cells. Can you please explain better such a choice? (Please  look at doi: 10.3389/fimmu.2022.883694)

Figure 5: please clarify what it Fluorescence intensity measurement? Can you please also provide data of the % of cells that have do degranulation.

To facilitate the reading and interpretation of the results, please replace the numbers by the names of the combinations. Same comment figure 6.

Can you please explain wht the specific lysis was not done for NK cells?

Figure 4, 5 and 6: Please indicate the number of replicates/donors used in each experiment.

Discussion:

Please link the discussion with the figure of the article, it will help the reader to follow it.

The discussion need to be improved to include a better analysis of the paper results and link the it with the known literature.

Line 464: Since the only cells that has been isolated are the NK and γδT cells and only those were compared. It is not possible to conclude it without the analysis of the type of cells in the PBMCs, as an effect due to CD8/CD4 T cells can not be excluded. (even though it was been described in the literature that CD8 do not reach to HSP70, please comment it and include it in the discussion).

Comments on the Quality of English Language

moderate Enghish editing required

Author Response

RESPONSE TO REVIEWER 1

The paper “Targeted delivery of HSP70 to tumor cells via supramolecular complex based on Her2-specific DARPin9_29 and the barnase:barstar pair” in cells is a article that  aims to demonstrate the efficiency in delivery HSP70 to the cell surface of the tumor cells (in this case HER2 expressing cells) using the technology Her2-specific DARPin9_29 and the barnase:barstar pair. This would allow the recognition through HSP70 and elimination of the tumor cells by the immune system. The article subject is of interest for the community, however I have several major concerns that lead me to refuse the paper for publication.  

Thank you; we comment further below.

it lacks the number of replicates of each experiment.

All treatments were performed in three replicates.

the characterization of the PBMCs before and after incubation with the target cells was not done

The characterization of the PBMCs before and after incubation with the target cells was not the scope of our study. The work was devoted only to assessing the antitumor effect of the two-module recombinant construct we developed.

Line 23 and below: please clarify it, as PBMCs also include B, DC cells which do not have a cytotoxic activity

Yes, PBMCs include different types of cells of the immune system, including those that do not have cytotoxic activity. However, the use of PBMCs in antitumor cytotoxic assays is one of the standard methods in experimental oncology research.

the importance of IL2 it is not clearly mentioned (Figure 4)

IL-2 increases the cytotoxic activity of NK cells as well as effector γδT cells.

Line 100: Please indicate the amount of cells?

We apologize for this omission.

We used 3 x 105 cells in 100 μl of PBA solution.

2.5 Paragraph: Please indicate the concentrations used?

The concentrations are given in Paragraph 2.4.

Please indicate what means PBA?

The composition of the buffer solution PBA is given in Paragraph 2.3.

The authors do not properly discuss for instance the differences observed between DARPin9_29-HSP70/ DARPin9_29-HSP70:barnase-HSP-70/16.

We include the following in the discussion:

The same is true when comparing in our delivery system the second modules containing either the full-length HSP70 protein or its C-terminal fragment with a molecular mass of 16 kDa. The maximum fluorescence intensity determined by flow cytometry and proportional to the number of bound molecules was observed for the 16 kDa fragment of HSP70. However, the full-length protein HSP70-labeled cells are more efficiently lysed by PBMC and isolated γδT lymphocytes (Figures 4 and 6).

Figure 1: Can you please explain the appearance of the other below bands, does it means that it is not pure or degraded?  How many milligrams could you obtained.

Yes, it is likely that the “lighter” bands that appear are the result of degradation of the HSP70 molecule.

The yield of recombinant protein DARPin9_29-HSP70 was 10 mg/L of growth medium as described in section Materials and Methods, Paragraph 2.2.

It would be easier for the reader to have the legend at the side of the histograms/ graphs. Considering that there is space to include the legends, please correct it.

We arranged all our figures and now it no need.

Please clarify what is fluorescence intensity (it is a FITC) in a and b.

We changed fluorescence intensity for FITC in histograms.

In c, by fluorescence intensity, you mean geometric mean? Please clarify it. Same comment for the below graph/histograms.

It was included in the figure legends.

Which statistic test was used, please indicate it here.

The paired Student’s t-test.

What means autofluorescence, it is from no stained cells?

Yes, it is true.

Paragraph 3.4: To better understand the effect of the combination in the induction of the cytotoxicity of the target cells, I strongly suggest that the cells population will be characterized (type of cells) for instance using specific antibody staining (e.g. CD20, CD3, CD14, CD16, CD45 etc) of the cells and analysed by flow cytometry.

Understanding the effect of the immune cell combination in the induction of the cytotoxicity for the target cells was not the scope of our study. The work was devoted only to assessing the antitumor effect of the two-module recombinant construct we developed.

Moreover, the authors should characterize the PBMCs upon stimulation with IL2, meaning type of cells that were expanded.

It is known that IL2 has activating effects on NK cells, as well as on T lymphocytes, including cytotoxic ones, therefore the addition of this cytokine to the culture medium is usually used in such cytotoxic tests.

I do not agreed in using this term as PBMCs as it includes several types if cells including non cytotoxic cells.

Yes, PBMCs include different types of cells of the immune system, including those that do not have cytotoxic activity. However, the use of PBMCs in antitumor cytotoxic assays is one of the standard methods in experimental oncology research. Moreover, it is important to specify PBMCs because the effects of specific cytotoxic effectors can be significantly influenced by surrounding cells, both through cytokines produced and through contact interactions.

Line 334: please indicate for how long they were incubated with IL2 before used as effector cells, and thus characterize the type of cells obtained after stimulation.

It was included in the text:

“At the initial stage, fractions of peripheral mononuclear cells (PBMC) isolated from the blood of healthy donors, were stimulated (overnight – 12 hours) with IL-2 and used as effectors”.

Figure 4: Consider the below comments. Please indicated/explained why those two ratios were done and better explain the difference observed between the different groups.

In the study, we used generally accepted effector/target cell ratios for this cytotoxic assay.

Line 352- it is not observed the same effect for the HSP70 alone?

Yes.

Paragraph 3.5/3.6: I could not fully understand why the authors decided to test the effect of these combinations using NK cells or γδT cells and nor for instance CD4, CD8 T cells. Can you please explain better such a choice? (Please  look at doi: 10.3389/fimmu.2022.883694)

Membrane-associated HSPs are usually detected on the surface of damaged, transformed and infected cells, in this case it is triggered a cascade of immunological reactions, including adaptive immune cells. We do not expect that tumor- and HSP-specific CD4 and CD8 T cells will be present in large numbers in healthy donors, although we agree that this possibility cannot be completely excluded. But the delivery system we developed is intended primarily to activate cytotoxic cells of the innate immune system.

Figure 5: please clarify what it Fluorescence intensity measurement? Can you please also provide data of the % of cells that have do degranulation.

We apologize for this mistake. It was corrected.

To facilitate the reading and interpretation of the results, please replace the numbers by the names of the combinations. Same comment figure 6.

We considered this possibility, but the legends were very voluminous for small histograms.

Can you please explain wht the specific lysis was not done for NK cells?

Our results indicate that NK cells have a high background level of cytotoxic activity against the target tumor cells we used. This may explain the lack of significant effect of HSP70 delivery to these cells. Additionally, this may be partly explained by the absence of other PMBC cells, which may play an accessory role in HSP70-induced cytotoxic NK cell activity.

Figure 4, 5 and 6: Please indicate the number of replicates/donors used in each experiment.

It was added to the Methods.

Discussion: Please link the discussion with the figure of the article, it will help the reader to follow it.

We added figure numbers into the discussion.

Line 464: Since the only cells that has been isolated are the NK and γδT cells and only those were compared. It is not possible to conclude it without the analysis of the type of cells in the PBMCs, as an effect due to CD8/CD4 T cells can not be excluded. (even though it was been described in the literature that CD8 do not reach to HSP70, please comment it and include it in the discussion).

We included the following in to discussion:

The results of the experiments using PBMCs as effector cells demonstrated a reliable anti-cancer effect of the treatment of target tumor cells with the two-module construct we developed. A small such effect was recorded in experiments with isolated from PBMCs cytotoxic γδT lymphocytes, and a similar trend was observed in experiments with NK cells isolated from peripheral blood. The latter can be explained by the high level of HSP70-independent cytotoxic activity of NK cells against the target tumor cells used. As for PBMCs, we hypothesize that T lymphocytes may have a cytotoxic effect on tumor cells with membrane-associated HSP70. At the same time, we do not exclude the involvement of antigen-presenting cells contained in PBMCs in the development of the lytic reaction of T-lymphocytes. Additionally, antigen-presenting cells can also enhance the cytotoxic response of NK cells by produced cytokines or/and by the interaction of classic and non-classic HLA class I molecules with the NK cell activating receptors.

Reviewer 2 Report

Comments and Suggestions for Authors

Here, the authors developed a darpin-based system for delivery of Hsp70 and its fragment to membranes of cancer cells. The work is sound and well done. My only concern is that the entire validation of the delivery system was done with cell culture. It is unclear if the in vitro effects of the Hsp70 delivery system on cytotoxicity of PBMC can be translated into real anti-cancer activities in an animal model.

A minor technical note - Fig 5 practically shows negative results. I suggest to combine Figs. 4, 5 and 6 into one mullti-panel figure.

Comments on the Quality of English Language

The English Language is fine.

Author Response

RESPONSE TO REVIEWER 2

Reviewer 2

Here, the authors developed a darpin-based system for delivery of Hsp70 and its fragment to membranes of cancer cells. The work is sound and well done. My only concern is that the entire validation of the delivery system was done with cell culture. It is unclear if the in vitro effects of the Hsp70 delivery system on cytotoxicity of PBMC can be translated into real anti-cancer activities in an animal model.

Thank you very much for your supportive comments.

Undoubtedly, this system should to be verified in animal experiments and we are going to do it in the future work.

A minor technical note - Fig 5 practically shows negative results. I suggest to combine Figs. 4, 5 and 6 into one multi-panel figure.

We arranged all the figures.

Reviewer 3 Report

Comments and Suggestions for Authors

This paper by Alekseeva, et al. describes a two-module system for delivering HSP70 to HER2-positive cells. The HER2-targeting domain, DARPin9_29, was fused to barnase while HSP70 or HSP16 was fused to barstar. The high affinity interaction between barnase and barstar enables HSP70/16 to efficiently bind to HER2-positive cells through association DARPin9_29. Interestingly, this two-module system achieved better HSP70 labeling efficiency than single molecule DARPin9_29-HSP70 fusion protein. HSP70-labeled cells are more efficiently lysed by PBMC than the control cells. This work is built upon a previous publication by the same author that used an scFv for HER2-targeting. Consequently, the novelty of this work is limited. It would be important for the authors to compare this new system with their previous construct and determine if there are any advantages. Below are some specific comments:

1.       What’s the concentration of DARPin9_29-HSP70 used in each experiment. The concentration of DARPin9_29-barnase and Barstar-HSP70/16 were 20 μg/mL and 50 μg/mL respectively (page 3, line 103-105). To properly compare the efficiency, it might be necessary to test DARPin9_29-HSP70 at both concentrations. I also suggest moving Figure S5 to the main text since it is the only quantitative data comparing the labeling efficiency of the one vs. two module systems.

2.       Page 11, line 436, the KD (Not Kd) should be ~10^-14 M (not 10-14 M). Similarly, line 437, the KD should be ~10^-15 (not 10-15 M)

3.       Considering the much better affinity between barnase and barstar, which is far superior to DARPin9_29 and Her2, it is surprising that a higher concentration of Barstar-HSP70 was used than DARPin9_29-barnase. Will a lower concentration of Barstar-HSP70/16 reduce background binding seen in Figure 3b?

Author Response

RESPONSE TO REVIEWER 3

This paper by Alekseeva, et al. describes a two-module system for delivering HSP70 to HER2-positive cells. The HER2-targeting domain, DARPin9_29, was fused to barnase while HSP70 or HSP16 was fused to barstar. The high affinity interaction between barnase and barstar enables HSP70/16 to efficiently bind to HER2-positive cells through association DARPin9_29. Interestingly, this two-module system achieved better HSP70 labeling efficiency than single molecule DARPin9_29-HSP70 fusion protein. HSP70-labeled cells are more efficiently lysed by PBMC than the control cells. This work is built upon a previous publication by the same author that used an scFv for HER2-targeting. Consequently, the novelty of this work is limited. It would be important for the authors to compare this new system with their previous construct and determine if there are any advantages. Below are some specific comments:

We thank you for your comments.

We did not aim to directly compare the effectiveness of our system for scFv and DARPin. Scaffold proteins and, in particular, darpins have many advantages over antibodies; they are more promising for the development of such artificial constructs.

What’s the concentration of DARPin9_29-HSP70 used in each experiment. The concentration of DARPin9_29-barnase and Barstar-HSP70/16 were 20 μg/mL and 50 μg/mL respectively (page 3, line 103-105). To properly compare the efficiency, it might be necessary to test DARPin9_29-HSP70 at both concentrations. I also suggest moving Figure S5 to the main text since it is the only quantitative data comparing the labeling efficiency of the one vs. two module systems.

The concentrations of the first and second modules used in the described experiments for treating target cells were selected based on a large series of previous studies.

We arranged all our figures.

Page 11, line 436, the KD (Not Kd) should be ~10^-14 M (not 10-14 M). Similarly, line 437, the KD should be ~10^-15 (not 10-15 M)

We apologize for these mistakes.

All designations have been corrected.

Considering the much better affinity between barnase and barstar, which is far superior to DARPin9_29 and Her2, it is surprising that a higher concentration of Barstar-HSP70 was used than DARPin9_29-barnase. Will a lower concentration of Barstar-HSP70/16 reduce background binding seen in Figure 3b?

The higher concentration of the second module is due to its size, which is significantly larger than the size of the first module, and the desire to reduce the difference in the molar concentrations of these modules. It is likely that the higher molar concentration of Barstar-HSP70/16 compared to Barstar-HSP70 was the reason for the difference in background binding in Figures 3a and 3b.

Round 2

Reviewer 1 Report

Comments and Suggestions for Authors

I consider that the authors reply to my comments and improve the paper accordingly. There is some disaccordance in some aspects in the paper, still I consider that the paper is now ready to be accepted. There are still minor corrections to be done, such as:

-  Figure 2, there are two scale bars in a, b, c

- In the Figure 3, 4 and 5, can you please indicate if there is statistical significance i.e. p values in the respective graphs. 

- supplementary figure S3, please increase the axes legend titles. 

Author Response

Thank you very much.

  1. We have deleted the extra elements in Figure 2.
  2. We included the following in the legends of Figures 3, 4 and 5: “The statistical significance (p<0.05 by Students’ t-test) is shown with the brackets.”
  3. We have enlarged the font size in the axis’s titles.

Reviewer 3 Report

Comments and Suggestions for Authors

My concerns have been addressed. However, I disagree that it is irrelevant to compare the current study with their previous study that used scFv. At least some discussion should be given to highlight the differences between them. 

Author Response

Thank you very much.

We included the following in the Discussion:

In our previous work [2], we used as a recognizing Her2-receptor module the scFv 4D5-dibarnase, which consists of two barnase molecules that are fused serially to the single-chain variable fragment (scFv) of humanized 4D5 antibody (30 kDa). The present work focuses on DARpin9_29 which is an artificial scaffold protein designed as alternative to antibodies. The advantages of this so-called alternative binding protein include its small size (18 kDa), which facilitates tumor penetration; the absence of Fc antibody- and complement-mediated cytotoxicity; high thermostability that enables its long-term storage at room temperature without loss of activity. Simplicity of production of DARpin9_29 fusion proteins make them promising for creating bispecific and multivalent constructs. Our confocal microscopy (Figure 2) and flow cytometry data (Figure 3) showed that DARPin9_29 in the dual-module system is as effective as the antibodies. The delivery of both the HSP70 and HSP70_16 molecules with DARPin9_29 also significantly enhanced the antitumor cytolytic effect of PBMC (Figure 4).